# Contribution of livestock H$_2$S to total sulfur emissions in a region with intensive animal production

Anders Feilberg [1], Michael Jørgen Hansen[1], Dezhao Liu[1,2] & Tavs Nyord[1]

Hydrogen sulfide (H$_2$S) from agricultural sources is generally not included in sulfur emission estimates even though H$_2$S is the major sulfur compound emitted from livestock production. Here we show that in a country with intensive livestock production (Denmark), agriculture constitute the most important sulfur source category (~49% of all sources of sulfur dioxide), exceeding both the production industry and energy categories. The analysis is based on measurements of H$_2$S using proton-transfer-reaction mass spectrometry. National emissions are obtained using ammonia as a reference pollutant with the validity of this approach documented by the high correlation of ammonia and hydrogen sulfide emissions. Finisher pig production is the most comprehensively characterized agricultural source of sulfur and is estimated to be the largest source of atmospheric sulfur in Denmark. The implication for other locations is discussed and the results imply that the understanding and modeling of atmospheric sulfate sources should include agricultural H$_2$S.

---

[1] Department of Engineering, Aarhus University, Hangøvej 2, 8200 Aarhus N, Denmark. [2] Present address: Zhejiang University, College of Biosystems Engineering and Food Science, 866 Yuhangtang Road, Hangzhou 310058, China. Correspondence and requests for materials should be addressed to A.F. (email: af@eng.au.dk)

Emissions of hydrogen sulfide ($H_2S$) contribute to the atmospheric burden of sulfur compounds, which have a major role in the formation of secondary aerosols through oxidation and conversion to aerosol sulfate[1, 2]. Aerosol sulfate is an important influence on earth radiation budget through reflection of sunlight and formation of clouds[3], and aerosol formation poses a threat to human health[4]. In general, the contribution of $H_2S$ has been considered to be of minor importance compared with sulfur dioxide ($SO_2$) from industry and fossil fuel combustion and dimethyl sulfide (DMS) from the marine biosphere[2, 5, 6].

The contribution of $H_2S$ to atmospheric sulfur is associated with large uncertainties. Sources of atmospheric $H_2S$ have been reported to be: oceans, wetlands, vegetation, salt marshes/estuaries, soil, tropical forests, and volcanoes[5–8], as well as a major anthropogenic contribution of 2.5% of $SO_2$ emissions from fossil fuel combustion[2, 7], which was estimated by Verma et al.[2] to be the major known source of $H_2S$. In a recent study on atmospheric sulfur particles, $H_2S$ was not included due to "the large uncertainties associated with its emission estimates"[9].

In the atmosphere, $H_2S$ reacts with OH radicals with a rate constant of $4.7 \times 10^{-12}$ cm$^3$ molecule$^{-1}$ s$^{-1}$ [10] corresponding to an estimated lifetime of 2.5 days. Gas phase reactions of $H_2S$ with $NO_3$ radicals[10] and ozone[11] are too slow to be considered important, but $H_2S$ react rapidly (as HS$^-$) with ozone in water droplets[12], which could represent an additional $H_2S$ sink despite its low solubility. The ultimate end-product of gas phase $H_2S$ oxidation in the atmosphere is considered to be $SO_2$[1], which in turn is oxidized and ends up as aerosol sulfate. Hence, the environmental effects of $H_2S$ emissions can be directly compared with the effects of $SO_2$ on a molar basis, but agricultural $H_2S$ is generally not considered as a source of secondary $SO_2$ in official estimates[13, 14]. The atmospheric lifetime of $SO_2$ has been estimated to be in the range of 4 to 48 h[15, 16], and together with the OH oxidation of $H_2S$ this means that $H_2S$ can be converted to aerosol sulfate on a relatively short timescale. $H_2S$ is being co-emitted with ammonia and organic amines from livestock production (including waste) with ammonia being emitted by far in the largest amounts[17–19]. This is important because ammonia and amines can enhance nucleation of $H_2SO_4$[20–22]. Thus, concurrent emission of $H_2S$ and ammonia/amines from livestock production facilities gives rise to a plume with a strong potential for aerosol formation.

Data on emissions of $H_2S$ from livestock production and waste is relatively scarce, but in recent years, the application of proton-transfer-reaction mass spectrometry (PTR-MS) has provided comprehensive datasets on $H_2S$ emissions[17, 23, 24] with detailed information on temporal variation.

In this work, emission of $H_2S$ in a region with intensive livestock production is estimated by using the concurrent emissions of ammonia ($NH_3$) as a reference pollutant. Denmark is used as a relevant case due to routinely reported emissions factors of $NH_3$[25–27], and due to its high density of livestock, comparable to northwestern Germany, Netherlands, Belgium, regions in Japan, Britany in France, Catalonia in Spain, states in USA (e.g., Iowa, North Carolina, Minnesota), and other regions with intensive livestock production.

The current paper presents data from measurement campaigns carried out over 6 years from 2009 to 2015. A part of the data was extracted from studies that were aimed at investigating odor and $NH_3$ emission abatement, and details concerning the locations and measurements can be found in these[17, 23, 24, 28]. The results clearly demonstrate that $H_2S$ from agriculture is a major source of atmospheric sulfur in Denmark and that agricultural $H_2S$ emissions from regions with intensive livestock production needs to be included in atmospheric sulfur budgets.

## Results

**Emission ratios of sulfur to nitrogen.** The results of a series of measuring campaigns are summarized in Table 1. As can be seen, the observed mass ratios of sulfur to nitrogen ($R_{S/N}$; g sulfur per g nitrogen (gS/gN)) lie within a relatively narrow range of $R_{S/N} = 0.10–0.26$ gS/gN for fattening pigs. In these calculations, only $H_2S$ has been considered since it is by far the most abundant sulfur compound. The only other sulfur compounds measured consistently in the ppb range are methanethiol and dimethyl sulfide, but these only constitute about 2–5% of $H_2S$. A summary of organic sulfur compound concentrations together with $H_2S$ data is presented in Table 2. In addition to methanethiol and dimethyl sulfide, mass-to-charge ratios ($m/z$) corresponding to dimethyl disulfide ($m/z$ 79 + 95) and dimethyl trisulfide ($m/z$ 127) were detected at very low levels of typically <1 and <0.1 ppb, respectively, and contributions of other compounds at these $m/z$ cannot be ruled out. Previously reported emissions[29–32] of dimethyl disulfide and dimethyl trisulfide should be disregarded due to their significant formation during sampling and analysis of

**Table 1 Overview of values of $R_{S/N}$ obtained from the data series included in the analysis**

| Test site and year | Animal category | $T_{out}$ (°C)[a] | $H_2S_{mean}$ (ppb) | $NH_{3mean}$ (ppm) | $R_{S/N}$ (gS/gN)[b] | $R^2$ | $n$ | No. of days |
|---|---|---|---|---|---|---|---|---|
| Site 1 (2009) | Pigs[c] | 12.2 | 265 | 3.7 | 0.15 (0.03–0.3) | 0.53 | 244 | 28 |
| Site 2A (2010) | Pigs | 13.3 | 373 | 7.3 | 0.18 (0.13–0.25) | 0.48 | 168 | 11 |
| Site 2B (2010) | Pigs | 15.4 | 301 | 4.7 | 0.10 (0.06–0.13) | 0.41 | 123 | 7 |
| Site 3A (2011) | Pigs | 11.7 | 511.3 | 7.4 | 0.24 (0.15–0.26) | 0.94 | 1307 | 54 |
| Site 3B (2011) | Pigs | 11.7 | 520 | 7.5 | 0.25 (0.15–0.26) | 0.87 | 1307 | 54 |
| Site 3 (2012) | Pigs | 8.6 | 420 | 4.3 | 0.23 (0.18–0.27) | 0.78 | 396 | 12 |
| Site 4 (2015) | Pigs | 11.9 | 348 | 3.0 | 0.26 (0.16–0.36) | 0.79 | 277 | 14 |
| Site 5 (2015) | Pigs | 5.6 | 259 | 3.7 | 0.14 (0.11–0.19) | 0.55 | 250 | 7 |
| Site 6A (Summer 2013) | Cattle | 18.5 | 133[d] | 4.6[d] | 0.12 (0.04–0.25) | 0.37 | 768 | 24 |
| Site 6B (Winter 2013) | Cattle | 0.4 | 9.1 | 2.4 | 0.009[e] (0.007–0.012) | 0.32 | 845 | 19 |

[a]The average outdoor temperature is included for comparison. The pig measurements were carried out in different seasons with little variation in average temperature and no significant effect of temperature on $R_{S/N}$ with the exception of cattle data (see text). The average outdoor temperature in Denmark is 8.5 °C
[b]The range is included in parentheses as the 5% and 95% percentiles
[c]All pig data are based on fattening pigs (30 to ~110 kg body mass)
[d]Weighted average of room and pit concentration
[e]Data only available for room air (containing 92% of the total emission; see "Methods" section for details)
$R_{S/N}$ is the mass ratio of sulfur to nitrogen emitted. Values of outdoor temperature and mean concentrations of $H_2S$ and $NH_3$ in parts-per-billion (ppb) and parts-per-million (ppm), respectively, are included for comparison together with the coefficients of determination ($R^2$) for the $H_2S$ versus $NH_3$ correlations

**Table 2 Composition of sulfur compounds emitted from pig production facilities and the contribution of organic sulfur compounds relative to H₂S**

| Location | H₂S (ppb) | Methanethiol (ppb) | Dimethyl sulfide (ppb) | $S_{org}/H_2S$ (%) |
|---|---|---|---|---|
| Site 1 | $265 \pm 255$ | $4.0 \pm 1.6$ | $4.1 \pm 3.4$ | 3.5 |
| Site 2 | $353 \pm 104$ | $12.0 \pm 3.4$ | $3.0 \pm 0.9$ | 4.2 |
| Site 3[a] | $468 \pm 290$ | $7.9 \pm 3.5$ | $3.0 \pm 5.2$ | 2.3 |
| Site 4 | $348 \pm 154$ | $4.6 \pm 2.9$ | $1.7 \pm 0.9$ | 1.8 |
| Site 5 | $259 \pm 68$ | $3.4 \pm 1.9$ | $3.5 \pm 1.1$ | 2.7 |

[a]Includes 2011 and 2012 data combined
Concentration ranges are provided as one standard deviation of the mean. $S_{org}/H_2S$ represents the sum of concentrations of methanethiol and dimethyl sulfide relative to the concentration of H₂S

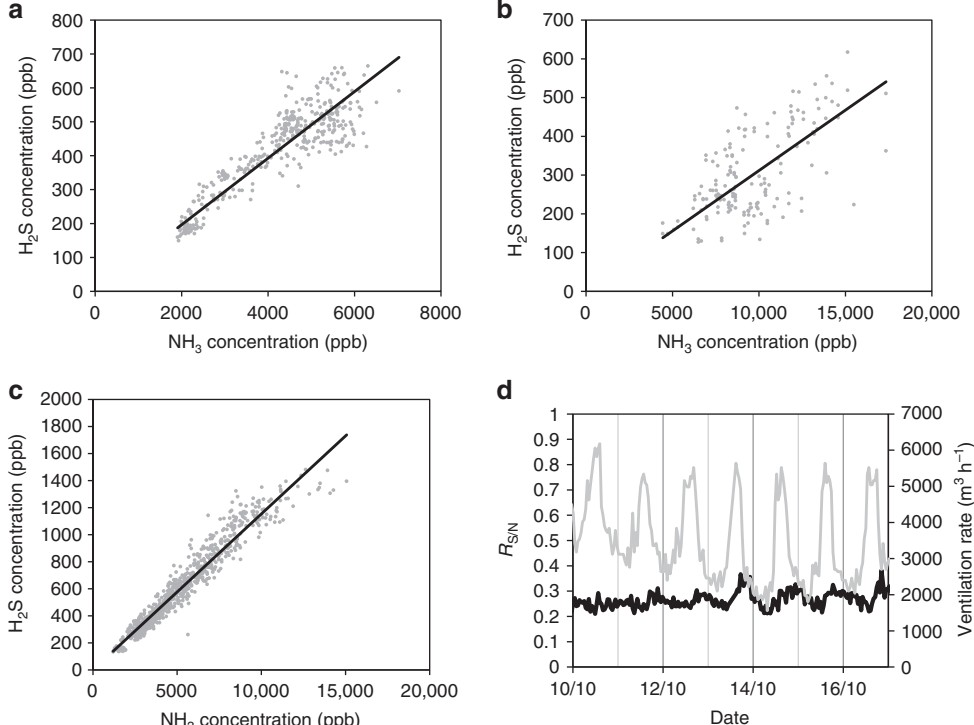

**Fig. 1** Examples of correlations between measured concentrations of H₂S and NH₃ in the ventilation outlet from two pig facilities. **a** Data from site 3 in 2012. **b** Data from site 2 in 2010. **c** Data from site 3 in 2011. **d** Temporal variation in the ratio of sulfur and nitrogen during 1 week of measurements (black line) together with temporal variation in ventilation rate with characteristic daytime maxima and nighttime minima (gray line). Full lines are least-squares linear regressions using a y axis intercept of 0. See Table 1 for further details

air-containing methanethiol when collecting samples for laboratory analysis[33, 34].

Generally, the emitted concentrations of H₂S and NH₃ are well correlated as seen in Table 1 and Fig. 1a–c. At site 3 (2011 data), measurements were performed on ventilation outlets from two identical pig units and the results were strikingly similar (Table 1). From the work reported here, it is observed that $R_{S/N}$ only varies moderately indicating that differences in compound properties are of minor importance. For example, $R_{S/N}$ varies surprisingly little with room ventilation rate as shown in Fig. 1d. It should be noted that ventilation rate is related to outdoor temperature to maintain a relatively constant temperature inside the pig facility, and no significant correlation of $R_{S/N}$ with $T_{out}$ was observed, in general. For individual data series, temperatures ranged in several cases from ~0 to ~25 °C and only in one case (site 3A–B, 2011), a significant temperature correlation was observed with lower $R_{S/N}$ at higher $T_{out}$ ($R^2 = 0.31$ and 0.66; data not shown) with a decrease in $R_{S/N}$ of 2% per °C. As $R_{S/N}$ was independent of $T_{out}$ for all other pig data, no attempts to normalize $R_{S/N}$ in relation to

$T_{out}$ was done. In any case, such a correction would be of little significance as the average outdoor temperature for all pig facility measurements was 11.2 °C, which is close to typical yearly average temperatures of 8–9 °C in Denmark.

For the cattle barn data, based on facilities with naturally driven ventilation, a clear difference between summer and winter is observed. During summer, $R_{S/N}$ is within the range of the pig house data and the inside temperature is also close to typical inside temperatures in pig houses. However, during winter the inside temperature is significantly colder than a pig house and was mostly between 6 and 10 °C.

In addition to animal houses, H₂S is emitted from liquid manure management as well, i.e., from manure storage and field application of manure. H₂S emission from manure storage is generally expected to be relatively low due to stagnant liquid conditions (limiting mass transfer) and the potential for surface oxidation[35]. From previous US data[36–39], an average $R_{S/N}$ value of 0.014 for pig manure storage can be inferred. As for manure application to fields, it has been observed recently that H₂S

**Table 3 Values of $R_{S/N}$ used for estimating sulfur emissions from agricultural sources**

| Emission source | NH$_3$ emission in Denmark (2011)$^a$ Gg N | $R_{S/N}$ (kg S per kg N) | Data source | Additional references |
|---|---|---|---|---|
| Pig houses | 12.8 | 0.19 ± 0.06$^b$ | This work | 18,39,41,42 |
| Cattle houses | 10.5 | 0.06 | This work | 41 |
| Pig manure storage | 1.7 | 0.014 | 37,39 | |
| Cattle manure storage | 1.5 | 0.04 | 53 | |
| Manure spreading (total) | 16.8 | 0.001 | This work | 40,51,52 |
| Poultry$^c$ | 1.8 | 0.01 | 41,54 | |
| Sheep and horses (total) | 0.8 | 0.01 | Not found$^d$ | |
| Fur (mink)$^c$ | 5.8 | 0.01 | Not found$^d$ | |

$^a$Data calculated based on information extracted from the Danish Normative System[27] and from published data from the Danish Centre of Environment and Energy[26]
$^b$One standard deviation included for pig data based on fattening pig values from Table 1
$^c$Housing and storage combined
$^d$Concurrent H$_2$S and NH$_3$ data not found. A conservative value of 0.01 is used, which is comparable to the lowest category (poultry). The contribution of sheep, horses and fur is estimated to be 2% of the total agricultural sources of H$_2$S
$R_{S/N}$ is the mass ratio of sulfur to nitrogen emitted from the source categories

emissions only occur in a short time frame after application[40], which suggests that H$_2$S emission from manure application is relatively low. On the basis of data extracted from a previous study[40], $R_{S/N}$ for manure application is estimated to be ~0.001, but this ratio is associated with considerable uncertainty, as the data were obtained under controlled conditions with one type of liquid manure. Following manure application, the manure surface is largely increased compared with manure storage and under these conditions surface oxidation of H$_2$S[35] will limit emission.

## Discussion

The values of $R_{S/N}$ are generally comparable with the limited literature data of simultaneous H$_2$S and NH$_3$ data both for pig and for cattle (Table 3). The only exception is nursery pigs for which higher ratios have been reported[39, 41], but it remains to be confirmed if this is typical. In general, the emissions of H$_2$S and NH$_3$ are well correlated and occur with a relatively constant ratio for each source supporting that the values of $R_{S/N}$ measured in this study can be extrapolated to regions with similar livestock production practices. For fattening pigs, $R_{S/N}$ is typically within a range of 0.1–0.25 gS/gN, whereas for cattle it appears to be lower, although more data are needed to confirm this. A significant correlation of H$_2$S and NH$_3$ emissions has previously been observed for finisher pigs[42].

For cattle, much lower $R_{S/N}$-values are observed at winter due to low H$_2$S concentrations, which indicate that production of H$_2$S from, e.g., sulfate reduction is strongly reduced at low temperature. Mass transfer rates of H$_2$S and NH$_3$ are not expected to be very differently influenced by temperature based on their diffusion coefficients and their enthalpy of liquid-to-air transfer[43]. The indoor temperature in pig buildings is typically controlled at 18–22 °C due to the mechanically driven ventilation and therefore much more constant throughout the year, than a cattle barn with natural ventilation.

In livestock facilities, both H$_2$S and NH$_3$ are primarily emitted from the liquid waste typically collected in manure pits under a slatted floor on which the animals reside[23]. Despite this common source, some variation in the ratio of H$_2$S to NH$_3$ would be expected for the following reasons: variation in pH of the slurry has opposite effects on the two compounds, as H$_2$S is a weak acid (p$K_a$ at 298 K is 7), whereas NH$_3$ is a base (p$K_a$ of NH$_4^+$ at 298 K is 9.25); variation in air turbulence above the emitting slurry surface is expected to affect NH$_3$ emission to a higher degree than H$_2$S emission, as mass transfer of NH$_3$ is mainly governed by air-side resistance[43]; H$_2$S emission is predominantly limited by liquid turbulence needed to increase transport of bulk liquid H$_2$S to the surface[43]. On the other hand, both H$_2$S and NH$_3$ originate mainly from protein in the feed. Nitrogen

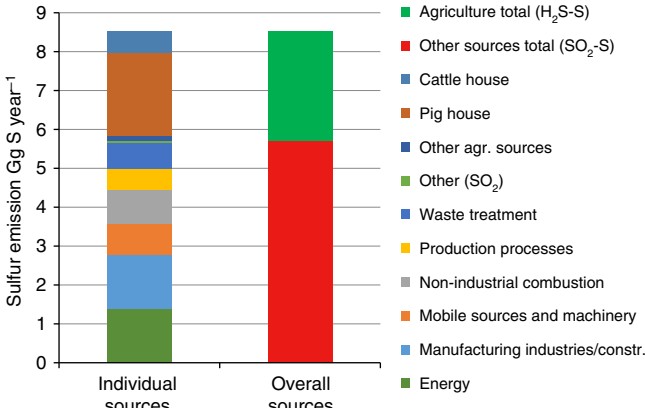

**Fig. 2** Estimated sulfur emissions for agricultural and non-agricultural sources. Sulfur is emitted as H$_2$S and SO$_2$ from agricultural and non-agricultural sources, respectively, and is reported in gigagrams sulfur per year. Data for non-agricultural sources (SO$_2$-S) is extracted from CEIP[13]. Other (SO$_2$) includes road transport, solvent use, and agricultural SO$_2$. Other Agricultural Sources include slurry application (all categories) as well as fur (mink), poultry, sheep, and horses. Sulfur emissions from agricultural sources (H$_2$S-S) are estimated based on values of $R_{S/N}$ measured as a part of this study or estimated based on available published data (Tables 1 and 3)

and sulfur are mainly excreted as urea and sulfate[44], which are converted to H$_2$S and NH$_3$ in the anaerobic manure slurry by ureolytic and sulfate-reducing bacteria. The relatively consistent values of $R_{S/N}$ and correlations of emissions suggest that the common source of H$_2$S and NH$_3$ outweighs the differences in compound characteristics.

For Denmark, detailed emission inventories for agriculture as well as other sectors have been available for a number of years. NH$_3$ emission inventories have been routinely updated by the Danish Centre for Environment and Energy as part of the unique Danish normative system[25–27]. This provides a strong basis for using NH$_3$ as a reference pollutant, which in combination with measured and estimated values of $R_{S/N}$ can provide the best available estimate of H$_2$S emissions from agriculture in Denmark. The result of this is provided in Fig. 2 using 2014 as a reference year based on officially reported inventories[13]. The total agricultural emission of sulfur (as H$_2$S) in Denmark is estimated to be 2.8 Gg S year$^{-1}$. Emission estimates for agricultural sources are compared with reported sulfur emission estimates from known sources[13, 45].

**Table 4 Comparison of sulfur emissions**

| Sources (DK and NL) | DK | NL | Sources (NC) | NC |
|---|---|---|---|---|
| SO$_2$-S[13]: | | | SO$_2$-S[14]: | |
| Combustion in energy and transformation industries | 1.38 | 9.30 | Fires | 0.65 |
| Non-industrial combustion plants (stationary sources) | 0.86 | 0.28 | Fuel combustion—comm/institutional | 1.80 |
| Combustion in manufacturing industry (stationary sources) | 1.38 | 4.34 | Fuel combustion—electricity generation (98% coal combustion) | 23.56 |
| Production processes (stationary sources) | 0.54 | 0.41 | Fuel combustion—industrial boilers | 3.63 |
| Solvent use and other product use | 0.02 | 0.00 | Fuel combustion—residential | 0.51 |
| Road transport | 0.04 | 0.09 | Industrial processes | 4.71 |
| Other mobile sources and machinery | 0.82 | 0.12 | Mobile sources | 0.58 |
| Waste treatment and disposal | 0.66 | 0.002 | Waste disposal | 0.11 |
| Agriculture (fossil fuel) | 0.01 | 0.00 | | |
| Total SO$_2$-S | 5.72 | 14.54 | Total SO$_2$-S | 36.8 |
| | | | | |
| H$_2$S-S: | | | H$_2$S-S: | |
| Pig houses | 2.13 | 2.95 | Pig houses | 5.73 |
| Cattle houses | 0.55 | 1.05 | Cattle houses | 0.15 |
| Other agricultural sources[a] | 0.12 | 0.19 | Other agricultural sources[a] | 1.01 |
| Total agricultural emissions | 2.81 | 4.19 | Total agricultural emissions | 6.90 |

[a]Waste storage, manure application, minor animal categories
Reported emissions of sulfur (SO$_2$-S) from known sources together with estimated agricultural emissions of sulfur (H$_2$S-S) in Gg per year for Denmark (DK), the Netherlands (NL) and the state of North Carolina (NC). Data is calculated for 2014 based on $R_{S/N}$ from the current study and reported emissions of NH$_3$

It is evident that livestock houses represent a significant source of atmospheric sulfur in Denmark and although uncertainties still remain, the agricultural contribution to sulfur emissions need to be accounted for. Pig housing is estimated to be the largest single source of atmospheric sulfur in Denmark with an emission of 2.13 Gg H$_2$S-S year$^{-1}$, which is higher than, e.g., the energy sector (1.38 Gg SO$_2$-S year$^{-1}$) or manufacturing industries (1.38 Gg SO$_2$-S year$^{-1}$). Fattening pigs is the agricultural source that is by far best characterized with consistent data. This livestock category is responsible for 68% of NH$_3$ emissions in Denmark and the equivalent H$_2$S contribution is estimated to be 76% with the notion that more data for sows and weaners are needed. More data for cattle production is needed, especially since a strong temperature variation is indicated by the data. Emissions from other agricultural sources are much more uncertain, but their combined contribution is estimated to be relatively small.

Uncertainties in the agricultural H$_2$S emission estimates are still expected to be considerable due to variation in farming practices, farm designs, manure handling systems, and feeding. Uncertainties in the reference NH$_3$ emission estimates as well as the variation in observed $R_{S/N}$ ratios (Table 1) influence these uncertainties. According to Mikkelsen et al.[26] uncertainties in NH$_3$ emission estimates for livestock buildings including pig houses are assessed to be 25%. Together with the variability in $R_{S/N}$ of pig houses (0.19 ± 0.06; Table 3) of 32%, this gives an uncertainty of 41% by error propagation. For all other H$_2$S source categories, much higher uncertainties are expected and more data is needed. However, pig houses are estimated to account for 76% of H$_2$S emissions from agriculture and if assuming that the uncertainty for all other sources is close to 100%, a propagated uncertainty (corresponding to one standard deviation) in total agricultural emissions of 40% is estimated. More knowledge about variability for different sources and climatic conditions is needed to verify this uncertainty and to clarify the variation in $R_{S/N}$.

The results presented here using Denmark as a case are expected to be general for similar animal production practices occur in other countries with intensive livestock production (Europe as well as regions in North America and Asia). The importance of H$_2$S-S relative to SO$_2$-S will of course depend on local conditions such as industrial production and fossil fuel combustion. To compare the results for Denmark with other locations, preliminary estimates based on $R_{S/N}$ are compared with

the Danish data in Table 4 for two cases, the Netherlands and North Carolina USA, based on 2014 data. These cases were selected as both are home to intensive livestock production and since relatively detailed emission data for NH$_3$ has been published[46, 47], which allows for application of the specific values of $R_{S/N}$. The importance of agricultural H$_2$S-S relative to SO$_2$-S is 19% in North Carolina, 29% in the Netherlands, and 49% in Denmark. In all three cases it is clear that agricultural H$_2$S is an important source of atmospheric sulfur that needs to be taken into account. In Denmark, a relatively high-livestock density together with low fossil fuel consumption gives rise to the highest influence of H$_2$S. In the Netherlands, combustion in energy and transformation industries is a relatively more important source compared with Denmark and contributes 64% of SO$_2$-S. However, agricultural H$_2$S emission is comparable to combustion in manufacturing industry in importance for the Netherlands and exceeds by far all other sources.

For North Carolina, the dominant source of atmospheric sulfur is coal combustion, but it should be noted that the strength of this source is rapidly declining and, for example, decreased from 41.3 Gg SO$_2$-S in 2011 to 23 Gg SO$_2$-S in 2014[14]. According to Table 4, the second largest source of sulfur in North Carolina is agriculture (emitted as H$_2$S) exceeding, e.g., industrial processes, fuel combustion in industrial boilers, and (by far) mobile sources. A previous estimate of agricultural H$_2$S emissions in North Carolina was attempted by Rumsey et al.[29] based on measurements at a single finisher pig facility. The statewide H$_2$S emission was estimated to be 1.2 Gg year$^{-1}$, which is considerably lower than the emission estimate in Table 4. No NH$_3$ data were provided for comparison, but measurements of both sulfur compounds and NH$_3$ were earlier performed at the same facility under similar conditions[42]. The livestock facility investigated[29, 42] is characterized by weekly discharges of manure in the housing system and this management practice will influence emissions. The NH$_3$ emission was 1.09 kg NH$_3$ animal$^{-1}$ year$^{-1}$ [42] (yearly average based on measurements in four seasons), whereas a value reported in a US meta-study for finisher pig production was 4.89 kg year$^{-1}$ animal$^{-1}$ (3.95 kg year$^{-1}$ animal$^{-1}$ for all pig categories). This shows that the specific facility used in the studies by both Blunden et al.[42] and Rumsey et al.[29] is not typical and that results from this facility should not be directly scaled by number of animals to achieve a statewide emission. Another

factor contributing to the relatively low H₂S emission estimate achieved by Rumsey et al.[29] is that weekly manure discharge in comparison to other manure management strategies is expected to influence $H_2S$ emissions to a larger degree than $NH_3$ emissions[39, 48]. Even though relatively low $H_2S$ emissions are expected from facilities with frequent manure discharge, the $R_{S/N}$-values extracted from Blunden et al.[42] are actually comparable to the values in Table 3 with the exception of summer conditions, which gives a low-value hinting to a potential influence of temperature. North Carolina summer temperatures are relatively high compared with, e.g., Denmark.

In general, the analysis presented here shows clearly that agricultural $H_2S$ is an important source of atmospheric sulfur and, hence, an important precursor for aerosol sulfate in the atmosphere. The data presented here is dominated by finisher pig production, which also appear to be the most important source. Other sources should be investigated more in depth and further data on, e.g., geographical distribution, the effects of temperature and the influence of management are needed to clarify further the significance of $H_2S$ emission from agriculture.

## Methods

**Measurement locations**. Emissions from pig farms have been measured at five different locations: An experimental pig production facility run under standard production conditions but with small pen sizes (site 1), a commercial full scale pig production facility (site 2), an experimental pig production facility run under standard production conditions with more typical pen sizes (site 3), and additional commercial full scale pig production facilities (site 4 and 5). At locations 2, 3, and 4, the data were obtained as part of testing of air scrubbers to treat emissions, but only untreated emission data are included here. At location 5, the data were obtained as part of testing manure treatment and only untreated emissions are included. No attempts were made to standardize production conditions, but all pig facilities are operated with typical feeding strategies (dry feeding ad libitum) and ventilation systems. Ventilation rate is controlled to maintain inside temperatures of 18–22 °C. All pig facilities were equipped with shallow manure pits (50–60 cm deep) used in all Danish pig facilities. These are discharged to the outside storage facility when full, typically at intervals of 5–6 weeks. Air exchange rates were obtained by using calibrated measuring fans (Fancom, the Netherlands) or a calibrated pressure difference method (Dynamic Air, SKOV, Denmark).

All emission data obtained for pig production were achieved by sampling exhaust air in the ventilation duct (outlet) of the pig facilities. Heated sampling lines (40–50 °C) were used to draw air samples to the instruments to minimize sampling-line adsorption. Sampling time varied from 10 to 20 min in each cycle. Background measurements were carried out in each measurement cycle using ambient air filtered with activated charcoal (Supelpure HC, Bellefonte, PA) and these were subtracted from the sample data.

One cattle farm (site 6) is included and measurements were done during both summertime and wintertime. This cattle farm was equipped with hybrid ventilation as detailed by Rong et al.[49] Hybrid ventilation is not typical of cattle barns but since the majority of the air exchange takes place in the naturally ventilated animal room, it is believed to be comparable to typical cattle barns. Air exchange in the naturally ventilated room was estimated by the standard method using $CO_2$ production from the animals as a naturally occurring tracer[50]. For the winter measurements, only data for the room air content of $H_2S$ and $NH_3$ was obtained. However, this was estimated to contain 92% of the emission in the summer and is assumed to have contained most of the emission in winter as well, although presumably a lower fraction than in summer. To use wintertime data, it was assumed that values of $R_{S/N}$ in the room and in the pit were comparable.

**Measurements by PTR-MS**. PTR-MS was used to monitor $H_2S$ concentrations as well as concentrations of ammonia ($NH_3$). A high-sensitivity quadropole PTR-MS (Ionicon, Austria) was used in all investigations. The PTR-MS was run at standard drift tube conditions with inlet and drift tube at 60–75 °C, drift tube pressure of 2.1–2.2 mbar, and a drift voltage of 600 V. This resulted in electrical charge-to-molecular densities in the range of 130–140 Townsend. The PTR-MS response to $H_2S$ ($m/z$ 35) was calibrated based on certified gas cylinders and by taking into account the dependency of the response to the sample air humidity as described previously[17, 24]. Other sulfur compounds, methanethiol ($m/z$ 49) and dimethyl sulfide ($m/z$ 63), were also measured by PTR-MS. The measurement of volatile sulfur compounds by PTR-MS (including calibration) is described in detail in previous papers[17, 24]. Dimethyl disulfide was measured by detection of $m/z$ 95 (M + 1) and $m/z$ 79 (fragment ion; loss of –CH₃). Owing to the contribution of phenol to $m/z$ 95, an upper limit of dimethyl disulfide was estimated on $m/z$ 79. Dimethyl trisulfide was measured by detection of $m/z$ 127.

An upper limit of $R_{S/N}$ for untreated manure application was estimated based on laboratory experiments with soil and manure in dynamic flux chambers. Details of the setup has been reported previously[40]. A concentration-time profile based on PTR-MS data was reconstructed and used with $NH_3$ data for this purpose, but the rapid cease in $H_2S$ emissions contributed to a significant uncertainty. Reliable $R_{S/N}$ data for manure application has not been obtained by other experiments, but field data confirm that $H_2S$ emission is low and ceases rapidly after application[51, 52].

$NH_3$ was at all locations measured by PTR-MS using $m/z$ 18 as the $NH_3$ signal and subtracting instrumental background at this mass-to-charge ratio. The instrumental background is relatively high for $m/z$ 18 due to ions formed in the ion source. For site 2 and 4, additional measurements were performed by photoacoustic IR detection using a factory-calibrated Innova photoacoustic analyzer 1412 and a multi-point sampler1309 (Lumasense Technology A/S, Denmark). In general, good agreement between PTR-MS and photoacoustic measurements were observed (differences typically within 10–20%) as has been reported previously[40].

**Data availability**. The datasets analyzed during the current study are available from the corresponding author on reasonable request.

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

## Acknowledgements

We thank the following for assistance with facilitating field measurements: A.L. Riis, K. Jonassen, A.P.S. Adamsen, L.B. Guldberg, E.F. Pedersen, A.L.T. Andersen. The work has been supported by the Danish Strategic Research Council (File Number 09-065200) and the Green Development Programme, GUDP (File Number 34009-13-0650).

## Author contributions

A.F. carried out measurements, performed data analysis, combined data from additional sources, and wrote the main part of the paper. M.J.H. and D.L. took part in performing measurements and carried out data treatment. T.N. was responsible for the setup for the liquid manure application test. All authors discussed the results and commented on the manuscript.

## Additional information

**Competing interests:** The authors declare no competing financial interests.

