## [Peer Review File · Nature Communications]

Reviewers' Comments:

Reviewer #1 (Remarks to the Author)

Review of the Nature Communications Manuscript # NCOMMS-16-28996-T

"Contribution of livestock emissions of H₂S to total sulfur emissions in a region with intensive livestock production" by Feilberg et al.

This manuscript attempts to examine hydrogen sulfide emissions and concentration from animal agricultural intensive livestock production region of Denmark.

The authors suggest that in a region with intensive livestock production, agriculture constitute the most important sulfur source category (~44% of all other sources combined), exceeding e.g. both the production industry and energy categories. The analysis is based on data collected during 2009 to 2015 using proton-transfer-reaction mass spectrometry. Emission estimates on a national level (Denmark) is obtained by using ammonia as a reference pollutant and the validity of this approach is documented by the generally high correlation of ammonia and hydrogen sulfide emissions. In addition, it is documented that hydrogen sulfide is the most significant sulfur compound with emissions by far exceeding organic sulfur compounds. Fattening pig production is the most comprehensively characterized agricultural source of sulfur and, in the investigated case of Denmark, this source is estimated to be the largest single source of atmospheric sulfur. The results imply that the understanding and modeling of atmospheric sulfate sources should be renewed.

The manuscript lacks novelty. Moreover, a number of the recent scientific manuscript relevant to this manuscript are not referenced/omitted. At places, the statement in the manuscript are misleading.

I cannot recommend publication of the manuscript in Nature Communications.

Major Comments:

1. Total sulfur budget includes SO₂, H₂S, DMS, DMDS, COS, etc. H₂S may be the largest sulfur compound emitted in Denmark (principally from animal waste) in animal agricultural regions as suggested by the authors. The authors suggest "Here we show that in a region (i.e. in Denmark) with intensive livestock production, agriculture constitute the most important sulfur source category (~44% of all other sources combined), exceeding e.g. both the production industry and energy categories." The statement/conclusion is misleading since, in general, in animal agricultural dominated region there are no other sulfur anthropogenic sources. While, this statement may be true for a selected region of Denmark, this conclusion is not correct for all animal agricultural producing countries of the world as SO₂ may be transported into the region being studied (e.g. as is the case for US Mid-West region). Therefore, the results of this study cannot be extrapolated to the rest of the world when it comes to the country as a whole. For example, Rumsey et al., (2014) indicate "H₂S swine CAFO emissions were estimated to constitute ~ 18% of North Carolina, USA, H₂S emissions" (Rumsey, I.C., V.P. Aneja, and W.A. Lonneman, 2014. "Characterizing reduced sulfur compounds emissions from a swine concentrated animal feeding operation", Atmospheric Environment, vol. 94, pp. 458-466). The authors have not cited this study.

2. The "emission estimates on a national level (Denmark) is obtained by using ammonia as a reference pollutant and the validity of this approach is documented by the generally high correlation of ammonia and hydrogen sulfide emissions." This is central to the authors argument in the manuscript. While this is correct, but this relationship was first observed by Blunden, J. et al., 2008 (Table 2) (Blunden et al., 2008, "Measurement and analysis of ammonia and hydrogen sulfide emissions from a mechanically ventilated swine confinement building in North Carolina", Atmospheric Environment, vol. 42, No. 14, pp. 3315-3331). This scientific paper too has not been cited.

The plot of the data (Table 2, Blunden et al., 2008) is as follows:

3. The authors state (line 129) "The results presented here using Denmark as a case are expected to be general for similar animal production practices occur in other countries with intensive livestock production (Europe as well as regions in North America and Asia)." I provide an approximate calculation to show that this final conclusion is incorrect as it relates to the US.

U.S. Total SO₂ emission 2014 (US Environmental Protection Agency, National Emissions Inventory (NEI, 2014)) $\sim 3 \times 10^6$ tons/year (does not include H₂S).

US H₂S pigs emissions 2012 $\sim 8,878$ tons/year

Source: Total number of pigs, 2012 (US USDA census) 66,026,787 pigs

H₂S emissions measured/calculated in North Carolina for barns plus lagoon for 10×10^6 pigs in NC 1.22×10^6 kg per year (Rumsey et al., 2014) (1 US ton = 907 kg).

Source: Rumsey, I.C., V.P. Aneja, and W.A. Lonneman, 2014. "Characterizing reduced sulfur compounds emissions from a swine concentrated animal feeding operation", *Atmospheric Environment*, vol. 94, pp. 458-466).

The calculation shows that, in the US, sulfur emission as H₂S from pigs is considerably smaller than sulfur emissions as SO₂ from the power sector.

Minor comments:

The authors should familiarize themselves with recent relevant scientific literature. Some references are provided:

1. Blunden et al., 2008, "Measurement and analysis of ammonia and hydrogen sulfide emissions from a mechanically ventilated swine confinement building in North Carolina", *Atmospheric Environment*, vol. 42, No. 14, pp. 3315-3331.
2. Blanes-Vidal, V.; Sommer, S. G.; Nadimi, E. S. Modelling surface pH and emissions of hydrogen sulphide, ammonia, acetic acid and carbon dioxide from a pig waste lagoon. *Biosyst. Eng.* 2009, 104, 510– 521.
3. Kim, K.Y., Ko, H.J., Kim, H.T., Kim, Y.S., Roh, Y.M., Lee, C.M., Kim, C.N., 2008. Quantification of ammonia and hydrogen sulfide emitted from pig buildings in Korea. *J. Environ. Manag.* 88, 195-202.
4. Rumsey, I.C., and V.P. Aneja, 2014. "Measurement and Modeling of Hydrogen Sulfide Lagoon Emissions from a Swine Concentrated Animal Feeding Operation", *Environmental Science & Technology*, vol. 48, pp. 1609–1617. [dx.doi.org/10.1021/es403716w](https://doi.org/10.1021/es403716w),
5. Rumsey, I.C., V.P. Aneja, and W.A. Lonneman, 2014. "Characterizing reduced sulfur compounds emissions from a swine concentrated animal feeding operation", *Atmospheric Environment*, vol. 94, pp. 458-466.

Reviewer #2 (Remarks to the Author)

This is a strong, timely, well executed and original piece of work. I commend the authors in identifying the issue, and excellent execution and follow through.

The measurements seem sound, and the use of NH₃ as a reference is both useful and allows a larger application of the results. This use of the H₂S correlation with NH₃, its relative constancy, and the subsequent use are the key assumptions in this paper.

However, although the correlation data is persuasive, it is clear that the assumption is clearest and strongest when there is a degree of temperature control (presumably because it is the anaerobes that are liberating the gases, and they are temperature dependent. It is also clear as the authors point out that R is not constant, but approximately so. This begs a question about the uncertainty on the calculated fluxes just based on variation of the ratio.

So three questions for the authors to think on:

a) you have scaled up your flux estimates based on the industries, but it is not clear to me how

you have extended your estimates beyond the very clear cut situation with reasonable temperatures $8 < T < 18$.

b) You give fluxes for total sector emissions...what have you counted and what have you not?

c) Some sort of uncertainty on the overall fluxes would be good

A nice piece of work, and a valuable addition.

Congratulations. Simon Watts.

Reviewer #3 (Remarks to the Author)

This paper reports estimations of H₂S emission from pig livestock production (and from one cattle) in Denmark. Those estimates were based upon proton-transfer-reaction mass spectrometry (PTR-MS) measurements of H₂S taken over a six-year period (from 2009 to 2015) and the concurrent measurement of NH₃. The author demonstrated nicely that the emissions of H₂S and NH₃ are closely correlated, and that the ratio RS/N (grams of sulfur per grams of nitrogen) can be used to estimate H₂S emissions in regions with similar livestock production practices based upon well characterized NH₃ emission inventories.

This paper reports interesting results and approach, and I support the publication of this paper in Nature Communication. However, there are few areas, discussed below, where further clarification and/or revisions could enhance the quality of the current manuscript.

- in the summary paragraph (line 15-16) "agriculture constitute the most important sulfur source category...": please clarify. Is this statement related to H₂S emissions exclusively? The importance of the different sectors may be very different if SO₂ sources were included, for example. It is a recurrent issue throughout the paper, and I would advise the authors to carefully check their use of 'sulfur source' or 'atmospheric sulfur'.

- in the summary paragraph (line 21-23), the authors claim that "it is documented that hydrogen sulfide is the most significant sulfur compound with emissions by far exceeding organic sulfur compounds". Although the claim may be right, there is only a very brief mention of respective measurements of organosulfur compounds and H₂S in the current text, on line 59-61. I think it would enhance the quality of the paper to have a supplementary table with those measurements to support the authors' claim, maybe in a Supplementary Information document.

- in the summary paragraph, line 25 "this source is estimated to be the largest single source of atmospheric sulfur": Do the authors mean here 'atmospheric reduced sulfur' (or 'atmospheric organosulfur')? Or is the intent to also include SO₂. If it is the latter, I do not think the statement is correct.

- in page 1, line 29-31: Although the emissions for H₂S are highly uncertain, could the author provide ranges of concentrations for each source for comparison with the pig livestock production reported in the present work?

- in page 2 line 35: should it read "...due to "the large uncertainties associated with its emissions estimates" "?

- My major concerns are associated with Table 1:

1) First, while there is a description for sites 1-4 (pig houses) and 6 (cattle), site 5 has no clear designation in the experimental section. Does site 5 corresponds to untreated manure application practices? Please clarify.

2) The third column in Table 1 represents outside temperatures at each site; however, in the main text, it is referred to the temperature inside the pig house a couple of times, without any data presented: on page 3, line 81 "during summer, RS/N is within the range of the pig house data and the inside temperature is close to typical inside temperatures in pig houses"; and again in page 3,

line 87 "the fattening pig data does not show any significant correlations between indoor temperature and RS/N". I would encourage the authors to add an additional column to Table 1 with inside temperature for each site to support those statements.

It isn't clear in the current manuscript if the measurements were made inside or outside the pig houses? From line 63 (and again on line 78 where it is referred to 'room ventilation'), it sounds like all measurements were performed inside the pig houses, with the exception of site 3 (2011). Is this correct? Is the transfer of emissions within the pig house to the outside atmosphere 100% efficient? Can the author comment on that?

3) The footnote (c) in Table 1 seems to be associated with the pig house data in the second column, second line; however, it also appears on the first line of the 6th site where the data correspond to cattle measurements. Please clarify.

4) It is fairly obvious from the difference of temperature observed for both site (site 6a and site 6b), but I would encourage the author to label site 6a 'summer' and site 6b 'winter' for clarity.

5) I do not understand the footnote (e) in Table 1. Please clarify.

- The RS/N values stated in the text (line 58; "RS/N = 0.09 – 0.27 gS/gN for fattening pigs") are different from Table 1 (RS/N = 0.10 – 0.26 gS/gN). Which one is the correct one?

- Figure 2: It could be misleading to label this figure "estimated sulfur emissions in gigagrams sulfur per year...". Those estimates reflect only the emissions of H₂S based on this work (and the defined RS/N values), and do not include any other organosulfur compounds (nor SO₂). It should read "estimated H₂S emissions in gigagrams sulfur per year..." instead.

- The same comment can be made for the sentence on line 117 "the total agricultural emission of sulfur is estimated to be 3.2 Gg S year⁻¹" - Again this corresponds to exclusively H₂S emissions.

- Methods section: Please provide information on the type of PTR-MS used; from previous studies from the same group (references 8, 17 and 18), I believe it was a quadrupole high sensitivity instrument. Please add appropriate details related to the current study.

- Methods section: If the above comment is correct, the instrument can only measure ions with nominal masses; in a previous study from Li and co-workers (Li et al., 2014), the authors discuss the possibility of an interference at m/z 35 originating from the isotopic ion [CH₃18OH + H]⁺ from methanol. Was methanol ever observed during the measurements? Was it an interference for the H₂S measurements?

Method section (line 228): The sentence "NH₃ was at all locations measured by PTR-MS using m/z 18 as the NH₃ signal and subtracting instrumental bias at this mass-to-charge ratio" isn't very specific. The background signal at m/z 18 have been attributed to the intrinsic formation of NH₃ in the ion source of the PTR-MS (Norman et al., 2007) and the background level is typically high for atmospheric measurements. From data presented in Table 1, the NH₃ levels encountered at the source, from pig houses, were always in the low ppm range, only 10 times higher than reported background (~100 ppb for a conventional PTR-MS). Can the author comment on what level of background concentration were observed for their studies, and whether or not this high background was ever a limitation for their measurements? Also, gas phase NH₃ is known to be a hard specie to measure due to its stickiness on sampling line wall or surface (Norman et al., 2007; Sintermann et al. 2011), did that affect the authors' measurements?

References cited:

1. Li, R.; Warneke, C.; Graus, M.; Field, R.; Geiger, F.; Veres, P. R.; Soltis, J.; Li, S. M.; Murphy, S. M.; Sweeney, C.; Petron, G.; Roberts, J. M.; de Gouw, J., Measurements of hydrogen sulfide (H₂S) using PTR-MS: calibration, humidity dependence, inter-comparison and results from field studies in an oil and gas production region. *Atmos Meas Tech* 2014, 7, (10), 3597-3610.
2. Norman, M.; Hansel, A.; Wisthaler, A., O₂(+) as reagent ion in the PTR-MS instrument: Detection of gas-phase ammonia. *Int J Mass Spectrom* 2007, 265, (2-3), 382-387.
3. Sintermann, J.; Sprigig, C.; Jordan, A.; Kuhn, U.; Amman, C.; Neftel, A., Eddy covariance flux measurements of ammonia by high temperature chemical ionisation mass spectrometry. *Atmos Meas Tech* 2011, 4, 599-616.

General response: The manuscript under consideration was initially submitted for Nature from which it was forwarded to Nature Communications. In the first place, we did not make any changes to the forwarded manuscript and the format for Nature was maintained. As a consequence, we had to exclude some details and also restrict the number of references to 30. This meant that we also had to exclude some relevant references as pointed out by the reviewers. In the revision, we will adhere to the Nature Communications format and include more details and references.

Reviewer 1.

Comment 1:

1. Total sulfur budget includes SO₂, H₂S, DMS, DMDS, COS, etc. H₂S may be the largest sulfur compound emitted in Denmark (principally from animal waste) in animal agricultural regions as suggested by the authors. The authors suggest “Here we show that in a region (i.e. in Denmark) with intensive livestock production, agriculture constitute the most important sulfur source category (~44% of all other sources combined), exceeding e.g. both the production industry and energy categories.” The statement/conclusion is misleading since, in general, in animal agricultural dominated region there are no other sulfur anthropogenic sources. While, this statement may be true for a selected region of Denmark, this conclusion is not correct for all animal agricultural producing countries of the world as SO₂ may be transported into the region being studied (e.g. as is the case for US Mid-West region). Therefore, the results of this study cannot be extrapolated to the rest of the world when it comes to the country as a whole. For example, Rumsey et al., (2014) indicate “H₂S swine CAFO emissions were estimated to constitute ~ 18% of North Carolina, USA, H₂S emissions” (Rumsey, I.C., V.P. Aneja, and W.A. Lonneman, 2014. “Characterizing reduced sulfur compounds emissions from a swine concentrated animal feeding operation”, Atmospheric Environment, vol. 94, pp. 458-466). The authors have not cited this study.

Response:

We are thankful for the reviewer for the opportunity to emphasize the relevance of our study for other areas than Denmark. First of all, we do not claim that our results can be extrapolated to the whole of US with a very diverse source composition and areas dominated by industry or agriculture or neither of the two. Our claim is however, that H₂S emissions from agriculture can be very important and so far overlooked on a *regional* scale, e.g. in states with relatively high livestock density relative to industrial sources. We agree with the reviewer that there may be different conditions and source strengths in other locations. Currently, however, agricultural sulfur emissions (as H₂S) are neglected as a source of SO₂ in official sulfur emission inventories both in the US and EU. This is important since the effects of sulfur emissions take place on a local and regional scale at which sulfur is oxidized during transport and forms sulfate particles. We believe

our results are very relevant for areas with high livestock density because we show that in such areas, where there may not be high SO₂ emissions from industry, there can still be very significant sulfur emissions from livestock production. We do not attempt to claim that H₂S is more important than SO₂ on a larger (global or continental) scale. However, we provide evidence that H₂S is an important and overlooked sulfur source, and therefore H₂S will adversely influence air quality in these areas. We would like to emphasize that we have included data for the whole country of Denmark and not only "a selected region" as claimed by the reviewer. Denmark is an industrialized country with a relatively dense population. We have carefully gone through the manuscript in order to clarify that our data should not necessarily be compared with e.g. all of US.

We have included a reference to Rumsey et al. in our revision and discussed our results in relation to this paper.

In our revision and with the possibility to add more text and references, we have made comparisons to two additional cases for which a reasonable amount of information is available, the Netherlands and North Carolina. We have done this to show the relevance of H₂S for other countries. In both cases we find that H₂S from livestock production is among the largest sources of sulfur. For Netherlands, a highly industrialized and densely populated country, livestock sulfur (H₂S) is exceeded by emissions of SO₂ from Combustion in Energy, but is comparable to Combustion in Industry (stationary). For North Carolina, we estimate that livestock H₂S is the second largest sulfur source in the state (using EPA NEI 2014 data) only exceeded by coal combustion, which still is a very dominating SO₂ source in the US, albeit declining over time. The additional data is included and discussed in the manuscript.

We obtain significantly higher H₂S emissions than Rumsey et al. and we accept that this may be puzzling to the reviewer. However, the estimate of Rumsey et al is only based on a single (and not particularly large hog farm). We would like to warn against such extrapolations, because a single farm may be managed in a way that does not allow emission data to be generalized. We believe that using NH₃ as a reference gas for which emissions is much better documented is a more robust approach because it takes into account different management practices and facility designs (as much as it is take into account for NH₃ emissions. The underlying assumption is that the ratio of S to N is relatively constant and this is exactly what we document in our manuscript both by measuring in a number of locations and by comparison to international data, including data from the US.

To illustrate that the H₂S data from Rumsey et al. should not be extrapolated to all of North Carolina, we can compare the emission data of NH₃ with typical numbers. It turns out, that the average NH₃ emission per animal reported by Blunden and Aneja (2008) was 1.09 kg/animal/year (yearly average). The H₂S data from Rumsey et al. is obtained at the exact same facility and under similar conditions in 2008. NH₃ data is not included in Rumsey et al. so we use the data from Blunden and Aneja (measured in 2005). If we use the 10 million pigs mentioned by the reviewer, this gives a total emission of NH₃ for North Carolina of 10.9 Gg/year. However, the total emission of NH₃ from "Livestock waste" in 2008 was 154.2 Gg/year. According to Stephen and Aneja (Atmos Environ 2008 vol 42 pp 3238) 47% of NH₃ emissions in North Carolina is due to hogs. If we assume

that 50% is from lagoons we arrive at a hog barn emission of 36.2 Gg/year for North Carolina, a factor of 3.3 higher than if one extrapolates data from a single barn. We can refer to Liu et al (2014; meta-study cited in our manuscript) for data on typical emissions of NH₃ from different housing categories. They arrive at an average number of 4.89 kg/animal/year for finisher pigs, much higher than the 1.09 reported by Blunden and Aneja. There can be several reasons why data from a single barn cannot be extrapolated, and it should be noted that this barn is equipped with a shallow pit with weekly manure discharge, which affects emissions and may not be typical for North Carolina, and that measurements were carried out in limited time periods. A shallow pit and frequent discharge is actually expected to affect H₂S emissions more than NH₃ emissions because different processes limits the flux of these two compounds; NH₃ flux is limited by air-side diffusion and H₂S is limited by liquid-side resistance. Shallow pit and frequent discharge may contribute to the relatively low H₂S emissions measured by Rumsey et al for this specific barn. This is also discussed by Liu et al (2014) stating that H₂S emissions are high for deep pits and decreases with increasing discharge frequency.

Comment 2:

2. The “emission estimates on a national level (Denmark) is obtained by using ammonia as a reference pollutant and the validity of this approach is documented by the generally high correlation of ammonia and hydrogen sulfide emissions.” This is central to the authors argument in the manuscript. While this is correct, but this relationship was first observed by Blunden, J. et al., 2008 (Table 2) (Blunden et al., 2008, “Measurement and analysis of ammonia and hydrogen sulfide emissions from a mechanically ventilated swine confinement building in North Carolina”, *Atmospheric Environment*, vol. 42, No. 14, pp. 3315-3331). This scientific paper too has not been cited.

The plot of the data (Table 2, Blunden et al., 2008) is as follows:

Response:

First, there was a simple mistake in the manuscript because we meant to cite two papers from the same authors and the same year; Blunden and Aneja (2008a and b). The reference software confused these, and one was cited in the wrong place (reference 33 in Table 2 of the original manuscript) and the other was not included, for which we apologize. It is now corrected and both are included. Although the graph provided by the reviewer was not presented in the Blunden and Aneja (2008) paper, we acknowledge that a positive correlation was observed between H₂S and NH₃ concentrations. This obviously supports our idea of using R_{S/N}. We are as mentioned including the paper in the revised manuscript (as was our intention all along). The novelty of our work is not so much the observation of correlation between H₂S and NH₃, but the observation that the slopes of such correlations are quite similar for e.g. pig production and comparable to S/N-ratios in other countries. This leads to the second and most important novelty that we can use this to estimate H₂S emissions by using NH₃ as a reference and hence obtain more robust emission estimates than if extrapolating from a single barn.

Comment 3:

3. *The authors state (line 129) “The results presented here using Denmark as a case are expected to be general for similar animal production practices occur in other countries with intensive livestock production (Europe as well as regions in North America and Asia).” I provide an approximate calculation to show that this final conclusion is incorrect as it relates to the US.*

*U.S. Total SO₂ emission 2014 (US Environmental Protection Agency, National Emissions Inventory (NEI, 2014)) ~3*10⁶ tons/year (does not include H₂S).*

US H₂S pigs emissions 2012 ~ 8,878 tons/year

Source: Total number of pigs, 2012 (US USDA census) 66,026,787 pigs

*H₂S emissions measured/calculated in North Carolina for barns plus lagoon for 10*10⁶ pigs in NC 1.22*10⁶ kg per year (Rumsey et al., 2014) (1 US ton = 907 kg).*

Source: Rumsey, I.C., V.P. Aneja, and W.A. Lonneman, 2014. “Characterizing reduced sulfur compounds emissions from a swine concentrated animal feeding operation”, Atmospheric Environment, vol. 94, pp. 458-466).

The calculation shows that, in the US, sulfur emission as H₂S from pigs is considerably smaller than sulfur emissions as SO₂ from the power sector.

Response

This comment has been addressed in the first response. Just to reiterate: We do not think that it is robust to extrapolate from a single observation series from one location and for a limited time. We have argued above that the emissions of both H₂S and NH₃ at the site studied in both Aneja et al (2008) and Rumsey et al 2014) are most likely not typical for hog production in North Carolina. Based on official NH₃ emission inventories and published data on different emission categories we use our R_{S/N} to estimate H₂S emissions from livestock production in North Carolina. We arrive at a number of 6900 tons of H₂S-Sulfur per year (2014) for North Carolina, obviously higher than the

1220 reported by Rumsey et al (by extrapolating from a single source). As stated in response #1, the comparison should be made for US states with high livestock production, not all of USA. We estimate that H₂S is the second largest sulfur source in North Carolina, exceeded only by Electricity Generation by Coal (NEI database), but higher than e.g. Fuel Combustion by Industrial Boilers and combined Industrial Processes. It is not too surprising that coal-derived SO₂ is more important in North Carolina than in Denmark, since electricity generation in Denmark is to a larger degree based on renewable energy incl. wind turbines. In the future one could expect that the situation will change in USA with less SO₂ from coal combustion and other fossil fuel sources.

A similar calculation have been made for the Netherlands and again we arrive at a high contribution by H₂S amounting to 29% of the total known land sources of SO₂ in terms of mass of sulfur. This is not far from the 49% calculated for Denmark. For North Carolina, the equivalent number is 19%. In all three cases, the current official emission inventories for agricultural sulfur are zero.

We have expanded our discussion to include these important findings and circumstances and have included comparisons of sulfur sources in the three cases presented, Denmark, the Netherlands and North Carolina.

Reviewer 2.

However, although the correlation data is persuasive, it is clear that the assumption is clearest and strongest when there is a degree of temperature control (presumably because it is the anerobes that are liberating the gases, and they are temperature dependent. It is also clear as the authors point out that R is not constant, but approximately so. This begs a question about the uncertainty on the calculated fluxes just based on variation of the ratio.

So three questions for the authors to think on:

a) you have scaled up your flux estimates based on the industries, but it is not clear to me how you have extended your estimates beyond the very clear cut situation with reasonable temperatures $8 < T < 18$.

b) You give fluxes for total sector emissions...what have you counted and what have you not?

c) Some sort of uncertainty on the overall fluxes would be good

a) Our measurements do not indicate any clear relation with outdoor temperature (except for cattle as addressed in the manuscript) within the temperature ranges investigated. It is correct that we determine R_{S/N} for moderate average temperatures, but the datasets actually covers larger temperature ranges. Several of the datasets covers temperatures from 0°C to ~25°C. There is not a consistent correlation of R_{S/N} with temperature in this temperature range although in one single case (Site 3, 2011) there is a relatively weak anti-correlation of R_{S/N} with temperature. Thus, even if the ratio at more extreme temperatures should be investigated, our data indicates that at typical temperatures of Denmark (0 to 25 °C; yearly average ~8-9 °C), the values determined in this study are representative and not clearly related to temperature. We have included remarks about this in the manuscript, but have chosen not to include further data on this matter since temperature appear to be of less significance.

b) We have included all known sources of SO₂ as listed by CEIP and by the Danish emission inventory (Mikkelsen et al. 2014). The categories of sources in the graphs and tables correspond either to CEIP or to US EPA (NEI) in the case of North Carolina (included in the revised version). The North Carolina data has been grouped in main categories covering all stated emissions of SO₂. We have not included international shipping as a source of SO₂. Undoubtedly, shipping is an additional important source of SO₂, but in this work, we want to compare with national ground-based sources. We have not included other sources of H₂S such as wastewater treatment plants as no official data has been found.

c) The following paragraph has been included regarding uncertainty:

“According to Mikkelsen et al. {Mikkelsen 2014} uncertainties in NH₃ emission estimates are assessed to be 25%. Together with the variability in R_{S/N} of pig houses (0.19±0.06; Table 3) of 32%, this gives an uncertainty of 40% by error propagation. For all other H₂S source categories, much higher uncertainties are expected and more data is needed. However, pig houses are estimated to account for 76% of H₂S emissions from agriculture and if assuming that the uncertainty for all other sources is close to 100%, a propagated uncertainty in total agricultural emissions of 47% is estimated.”

Reviewer 3.

- in the summary paragraph (line 15-16) “agriculture constitute the most important sulfur source category...”: please clarify. Is this statement related to H₂S emissions exclusively? The importance of the different sectors may be very different if SO₂ sources were included, for example. It is a recurrent issue throughout the paper, and I would advise the authors to carefully check their use of ‘sulfur source’ or ‘atmospheric sulfur’.

We have revised the manuscript according to the comment and have indicated sulfur of different origin by labeling these as H₂S-S and SO₂-S. We have made the comparison on the basis of sulfur mass, since H₂S and SO₂ are equivalent in the sense that H₂S is rapidly oxidized to SO₂ in the atmosphere with a lifetime of 2.5 days. The comparison on a sulfur mass basis shows that agricultural emission of H₂S-S in the case of DK exceeds the single source categories of SO₂-S, but not all SO₂-S of which H₂S-S constitute close to 50%. This is mentioned in the revised manuscript.

- in the summary paragraph (line 21-23), the authors claim that “it is documented that hydrogen sulfide is the most significant sulfur compound with emissions by far exceeding organic sulfur compounds”. Although the claim may be right, there is only a very brief mention of respective measurements of organosulfur compounds and H₂S in the current text, on line 59-61. I think it would enhance the quality of the paper to have a supplementary table with those measurements to support the authors’ claim, maybe in a Supplementary Information document.

This is indeed a relevant comment and we have included a small table with contributions of organic sulfur compound occurrence together with H₂S. The observations are discussed in the revised text.

- in the summary paragraph, line 25 “this source is estimated to be the largest single source of atmospheric sulfur”: Do the authors mean here ‘atmospheric reduced sulfur’ (or ‘atmospheric organosulfur’)? Or is the intent to also include SO₂. If it is the latter, I do not think the statement is correct.

We understand the confusion from the phrasing. As mentioned in the reply to the first comment, we estimate that agricultural H₂S-S corresponds to close to 50% of all ground-based sources of SO₂-S in Denmark. However, comparing to individual source categories (Table 4), agricultural H₂S-S (dominated by pig production) exceeds the SO₂-S sources. This is clarified in the revised manuscript.

- in page 1, line 29-31: Although the emissions for H₂S are highly uncertain, could the author provide ranges of concentrations for each source for comparison with the pig livestock production reported in the present work?

We have looked into this, but we do not think that concentration ranges is a good indicator for comparing emission sources (flux). We have not found any other meaningful way to compare emissions strengths on a global scale (from the references) to national livestock emission data.

- in page 2 line 35: should it read “...due to “the large uncertainties associated with its emissions estimates” “?

Yes. This has been corrected.

- My major concerns are associated with Table 1:

1) First, while there is a description for sites 1-4 (pig houses) and 6 (cattle), site 5 has no clear designation in the experimental section. Does site 5 corresponds to untreated manure application practices? Please clarify.

The description of site 5 was omitted by mistake. It is included in the Methods section.

2) The third column in Table 1 represents outside temperatures at each site; however, in the main text, it is referred to the temperature inside the pig house a couple of times, without any data presented: on page 3, line 81 “during summer, RS/N is within the range of the pig house data and the inside temperature is close to typical inside temperatures in pig houses”; and again in page 3, line 87 “the fattening pig data does not show any significant correlations between indoor temperature and RS/N”. I would encourage the authors to add an additional column to Table 1 with inside temperature for each site to support those statements.

Inside temperatures in pig houses are typically controlled to be at 18-20°C. As mentioned in the response to Reviewer 2, this means that ventilation rate is linked to outside temperature. We have clarified this in the Methods section of the revised manuscript, but we do not think it adds much value to include inside temperatures in Table 1 as they are within a relatively narrow range.

It isn't clear in the current manuscript if the measurements were made inside or outside the pig

houses? From line 63 (and again on line 78 where it is referred to 'room ventilation'), it sounds like all measurements were performed inside the pig houses, with the exception of site 3 (2011). Is this correct? Is the transfer of emissions within the pig house to the outside atmosphere 100% efficient? Can the author comment on that?

This is indeed a relevant comment. All measurements in pig houses were done in the ventilation duct in order to ensure that representative emission data is obtained. This is clarified in the Methods section.

3) The footnote (c) in Table 1 seems to be associated with the pig house data in the second column, second line; however, it also appears on the first line of the 6th site where the data correspond to cattle measurements. Please clarify.

The second "c" should have been a "d". This has been corrected. Thanks for finding this mistake.

4) It is fairly obvious from the difference of temperature observed for both site (site 6a and site 6b), but I would encourage the author to label site 6a 'summer' and site 6b 'winter' for clarity.

We have followed this recommendation.

5) I do not understand the footnote (e) in Table 1. Please clarify.

The explanation is provided in the Methods section. We have referred to this in the revised footnote.

- The RS/N values stated in the text (line 58; "RS/N = 0.09 – 0.27 gS/gN for fattening pigs") are different from Table 1 (RS/N = 0.10 – 0.26 gS/gN). Which one is the correct one?

The numbers in the table are the correct ones. It has been corrected in the text.

- Figure 2: It could be misleading to label this figure "estimated sulfur emissions in gigagrams sulfur per year...". Those estimates reflect only the emissions of H₂S based on this work (and the defined RS/N values), and do not include any other organosulfur compounds (nor SO₂). It should read "estimated H₂S emissions in gigagrams sulfur per year..." instead.

The figure actually includes both H₂S-S from agriculture and SO₂-S from other sources. We have clarified this in the figure legend as well as throughout the text.

- The same comment can be made for the sentence on line 117 "the total agricultural emission of sulfur is estimated to be 3.2 Gg S year⁻¹" - Again this corresponds to exclusively H₂S emissions. We have clarified this by using H₂S-S.

- Methods section: Please provide information on the type of PTR-MS used; from previous studies from the same group (references 8, 17 and 18), I believe it was a quadrupole high sensitivity instrument. Please add appropriate details related to the current study.

Details have been added.

- Methods section: If the above comment is correct, the instrument can only measure ions with nominal masses; in a previous study from Li and co-workers (Li et al., 2014), the authors discuss the possibility of an interference at m/z 35 originating from the isotopic ion [CH₃¹⁸O⁺H] from methanol. Was methanol ever observed during the measurements? Was it an interference for the H₂S measurements?

We have measured methanol (m/z 33) on most occasions. Methanol is present at levels of ~10 ppb in pig house emissions. Since the ¹⁸O abundance is 0.2%, the potential interference of methanol on H₂S is <1%.

Method section (line 228): The sentence “NH₃ was at all locations measured by PTR-MS using m/z 18 as the NH₃ signal and subtracting instrumental bias at this mass-to-charge ratio” isn’t very specific. The background signal at m/z 18 have been attributed to the intrinsic formation of NH₃ in the ion source of the PTR-MS (Norman et al., 2007) and the background level is typically high for atmospheric measurements. From data presented in Table 1, the NH₃ levels encountered at the source, from pig houses, were always in the low ppm range, only 10 times higher than reported background (~100 ppb for a conventional PTR-MS). Can the author comment on what level of background concentration were observed for their studies, and whether or not this high background was ever a limitation for their measurements? Also, gas phase NH₃ is known to be a hard specie to measure due to its stickiness on sampling line wall or surface (Norman et al., 2007; Sintermann et al. 2011), did that affect the authors’ measurements?

This is a very relevant comment. We have added a paragraph on sampling to the Methods section to explain this.

We typically measured background m/z 18 signals corresponding to ~500 ppb and subtracted these from the sampling data. We included background measurements in each measurement cycle (i.e. 1-2 measurements per hour) and obtained these by measuring the m/z 18 (and other masses) using filtration by activated charcoal. The measured levels of NH₃ were indeed in the order of 10 times higher than the background, but since background was subtracted for each measurement cycle, this background contribution by the ion source is accounted for. It should also be noted that despite the relatively high background, the method detection limits for NH₃ was typically <50 ppb (calculated from the variation/“noise” of the background signals). Thus our measurements were well above the detection limits of PTRMS for NH₃.

In relation to sampling line adsorption, we used heated sampling lines (40-50°C) and measured for sufficiently long time to ensure that signals were stable (10-20 minutes) even for NH₃. This information is also included in the Methods section.

References cited:

1. Li, R.; Warneke, C.; Graus, M.; Field, R.; Geiger, F.; Veres, P. R.; Soltis, J.; Li, S. M.; Murphy, S. M.; Sweeney, C.; Petron, G.; Roberts, J. M.; de Gouw, J., Measurements of hydrogen sulfide (H₂S) using PTR-MS: calibration, humidity dependence, inter-comparison and results from field

- studies in an oil and gas production region. *Atmos Meas Tech* 2014, 7, (10), 3597-3610.
2. Norman, M.; Hansel, A.; Wisthaler, A., O-2(+) as reagent ion in the PTR-MS instrument: Detection of gas-phase ammonia. *Int J Mass Spectrom* 2007, 265, (2-3), 382-387.
 3. Sintermann, J.; Sprigig, C.; Jordan, A.; Kuhn, U.; Amman, C.; Neftel, A., Eddy covariance flux measurements of ammonia by high temperature chemical ionisation mass spectrometry. *Atmos Meas Tech* 2011, 4, 599-616.

Reviewers' Comments:

Reviewer #1:

Remarks to the Author:

The manuscript is much improved, and addresses my comments and suggestions. I recommend that the manuscript be published after my suggestions are comments are addressed.

1. line 154 and 155: " e.g. the Energy sector (1.38 Gg H₂S-S year⁻¹) or Manufacturing Industries (1.38 Gg H₂S-S year⁻¹)." Do the authors mean SO₂-S, or H₂S from these sectors were reported.
2. Line 180: Add USA after North Carolina.

Reviewer #2:

Remarks to the Author:

It seems to me that the main point of this work is to make the point that agricultural S emissions are not included in emissions inventories, but that in places with reasonable amounts of agriculture, hence high H₂S emissions, these proxy-SO₂ emissions are significant and hence probably should be. My view is the authors have generated robust data, and have made that case.

It seems to me this work will be of interest to others in the field, and given the measurements methodology and the fact they span 6 of 7 years, the results are convincing and seem robust.

Reviewer #3:

Remarks to the Author:

The authors demonstrated efforts in revising the manuscript, and as a result, the quality is greatly enhanced. The paper is much clearer and easy to follow. I believe all the comments and points raised by the different reviewers have been satisfactorily addressed, and I recommend this manuscript for publication in Nature Communication.

Please find below a few editorial notes to address prior the publication:

- The last table, entitled "Reported emissions of sulfur (SO₂-S) from known sources together with estimated agricultural emissions of sulfur (H₂S-S) in Gg per year for Denmark (DK), the Netherlands (NL) and the state of North Carolina (NC). Data for 2014." should be Table 4, not Table 3.

- In addition, reference to this table should be changed in the main text (line 180, line 193, line 198, and line 212).

- Lastly, on Table 4, I would encourage the authors to add a sentence in the Table caption or a footnote to reiterate here that the HS₂-S values are coming from RS/N values from reported NH₃ emissions. It is nicely mentioned in the main text, but a reader who would only look at Table 4 would miss the information entirely.

- one line 244, I suggest to add "(site 6)" after "One cattle farm (site 6)..." for clarity.

A few minor corrections have been made according to the reviewer suggestions:

Reviewer 1:

1. line 154 and 155: " e.g. the Energy sector (1.38 Gg H₂S-S year⁻¹) or Manufacturing Industries (1.38 Gg H₂S-S year⁻¹)." Do the authors mean SO₂-S, or H₂S from these sectors were reported.

Response: This should be SO₂-S and has been corrected. Thanks.

2. Line 180: Add USA after North Carolina.

Response: This has been added.

Reviewer 2: No corrections required.

Reviewer 3:

- The last table, entitled "Reported emissions of sulfur (SO₂-S) from known sources together with estimated agricultural emissions of sulfur (H₂S-S) in Gg per year for Denmark (DK), the Netherlands (NL) and the state of North Carolina (NC). Data for 2014." should be Table 4, not Table 3.

Response: This has been corrected.

- In addition, reference to this table should be changed in the main text (line 180, line 193, line 198, and line 212).

Response: This has been corrected; however, the last reference (original line 212) should still be made to Table 3.

- Lastly, on Table 4, I would encourage the authors to add a sentence in the Table caption or a footnote to reiterate here that the H₂S-S values are coming from RS/N values from reported NH₃ emissions. It is nicely mentioned in the main text, but a reader who would only look at Table 4 would miss the information entirely.

Response: This has been added accordingly.

- one line 244, I suggest to add "(site 6)" after "One cattle farm (site 6)..." for clarity.

Response: Good point. This has been added.